# Effects of Sulfamethoxazole on Fertilization and Embryo Development in the *Arbacia lixula* Sea Urchin

**DOI:** 10.3390/ani12182483

**Published:** 2022-09-19

**Authors:** Valentina Lazzara, Manuela Mauro, Monica Celi, Gaetano Cammilleri, Aiti Vizzini, Claudio Luparello, Paola Bellini, Vincenzo Ferrantelli, Mirella Vazzana

**Affiliations:** 1Dipartimento di Scienze e Tecnologie Biologiche, Chimiche e Farmaceutiche (STEBICEF), Università di Palermo, 90128 Palermo, Italy; 2Istituto Zooprofilattico Sperimentale Della Sicilia A. Mirri, 90129 Palermo, Italy

**Keywords:** antibiotic, echinoderms, embryos, environmental toxicity, gametes, invertebrates

## Abstract

**Simple Summary:**

Drugs released into the aquatic environment create serious problems for the organisms that live there. For this reason, the present study investigates the in vitro effects of the antibiotic sulfamethoxazole, widely found in wastewater, on the fertilization and development of the *Arbacia lixula* sea urchin. The results showed a significant reduction in the percentage of fertilized oocytes at the highest drug concentrations, together with an increase in anomalies and delays in the development of the embryo. Therefore, the data obtained suggest urgent intervention on the release of these drugs in order to prevent important alterations in the species’ development and to preserve biodiversity.

**Abstract:**

To date, drugs released into the aquatic environment are a real problem, and among antibiotics, sulfamethoxazole is the one most widely found in wastewater; thus, the evaluation of its toxicity on marine organisms is very important. This study, for the first time, investigates the in vitro effects of 4 concentrations of sulfamethoxazole (0.05 mg/L, 0.5 mg/L, 5 mg/L, 50 mg/L) on the fertilization and development of the sea urchin *Arbacia lixula*. The gametes were exposed to drugs in three different stages: simultaneously with, prior to, and post-fertilization. The results show a significant reduction in the percentage of fertilized oocytes at the highest drug concentrations. Moreover, an increase in anomalies and delays in embryo development following the treatment with the drug was demonstrated. Therefore, the data suggest that this antibiotic can alter the development of marine organisms, making it urgent to act to reduce their release and to determine the concentration range with the greatest impact.

## 1. Introduction

To date, different types of pollutants are released into aquatic environments, negatively affecting the lives of the aquatic organisms that inhabit them [1,2,3]. These organisms are very important as a resource of bioactive molecules as food, and in the study of immune responses under stress conditions and human disease [4,5,6,7,8]. The use of drugs is widespread in the treatment of human, animal, and plant diseases, and the problem of pharmaceutical waste being released into aquatic environments is becoming urgent [9]. After their use, the remaining drugs are usually disposed of in an inadequate manner via the sinks, toilets, and drains of hospitals and pharmaceutical factories, often in the form of active compounds that have not been metabolized, thus maintaining their original chemical structures and biological functions [10,11,12,13,14]. Conventional wastewater treatment plants often cannot effectively remove these pharmaceutical compounds (PCs) since their elimination is influenced by the characteristics of drugs, the techniques used, and the environmental conditions [15,16]. This leads to the frequent detection of PCs in wastewater, and their concentrations generally range from ng/L to μg/L [17,18,19]. It is known that their persistence in wastewater can negatively affect the ecosystem as these compounds are potentially toxic for non-target organisms, including marine animals [20,21]. In fact, several studies in the literature highlight the negative effects of drugs on different aquatic species, including oxidative stress, metabolic and immunological alterations, and reductions in the success of fertilization and the development of organisms [3,22,23,24,25,26,27]. Fertilization, in particular, is a critical phase of the lifecycle that ensures the preservation of a species, and the presence of pharmaceutical compounds can seriously interfere with this process, affecting the production and quality of the gametes. Indeed, De Oliveira et al. [23] reported on the adverse effects of 4 common pharmaceuticals (acetaminophen (1.2–9.0 mg/L), chlorpromazine (0.50–3.14 mg/L), diclofenac sodium (52.0–155.3 mg/L), and propranolol (4.0–11.9 mg/L) on the reproduction of *Daphnia magna* after analyzing the mobility of the gametes and the quantity of offspring. The authors demonstrated the dose-dependent immobility of gametes in the presence of all of the tested drugs, as well as a significant decrease in the production of offspring caused by chlorpromazine and propranolol. Similarly, another study on *Mytilus galloprovincialis* showed that diclofenac (100 μg/L) causes the down-modulation of tyrosine metabolism and the up-modulation of tryptophan metabolism, which are involved in the production and release of gametes [28]. Among these pharmaceuticals products, and specifically among the classes of antibiotics, sulfonamides—synthetic antimicrobial molecules whose structure mimes that of para-aminobenzoic acid (PABA)—are widespread [29,30]. One of the most common and diffuse drugs belonging to this category is sulfamethoxazole (SMX), a competitive inhibitor of the enzyme dihydropteroate synthase, which prevents the formation of dihydropteroic acid, a precursor of folic acid, which is required for bacterial growth. It is widely used in the treatment of several bacterial infections in humans, and it is effective against both Gram-positive and Gram-negative bacteria [31]. The average concentrations of the drug that are detected in wastewater range from ng/L to μg/L [32,33], and several studies have demonstrated its negative effects (i.e., oxidative stress, the suppression of the immune response, inflammation, the inhibition of acetylcholinesterase activity, the alteration of osmotic regulation, and the alteration of energy metabolism) in marine vertebrates and invertebrates [34,35,36,37]. However, other authors have recently reported that the concentrations of this drug discarded by the livestock industry and in aquaculture wastewater can rise to the level of mg/L [38,39]. Focusing on invertebrates, it is known that they play an important role as a source of bioactive molecules with antioxidant, anti-inflammatory, and anti-tumor effects [4,5,40,41,42], and they also represent useful bioindicators for the study of environmental conditions [43,44,45,46,47,48]. Among invertebrates, echinoderms possess peculiar characteristics which make them particularly suitable for applications in the ecotoxicological field, such as their wide distribution, the ease of sampling/maintaining them, the ease of collecting their gametes, their availability throughout the year, their external fertilization mechanism, their rapid development, and their sensitivity to a wide spectrum of pollutants [49]. Furthermore, their lifecycle stages, such as fertilization and embryonic division, are particularly sensitive to environmental conditions and are therefore increasingly used in the ecotoxicological approach to assess the quality of an environment [50,51]. In light of all of this, given the importance of these invertebrates as bioindicators, and considering the SMX concentrations reported by other authors [32,33,38,39], our study aimed to investigate for the first time the in vitro effects of exposure to sulfamethoxazole at different concentrations and at three different stages (simultaneously with, prior to, and post-fertilization) on *Arbacia*
*lixula* gametes and embryos. Our goal was to evaluate whether the drug negatively affects the gametes and embryos and whether this depends on the concentration. In fact, the obtained results could provide useful information for the regulation of sulfamethoxazole releases into water, thus protecting the species that live there.

## 2. Materials and Methods

### 2.1. Experimental Animals

A total of 54 specimens of *Arbacia lixula* (16 ± 1 g) were collected in the spring from the rocky seabed of Mongerbino (Gulf of Palermo) and housed in the aquarium of the University of Palermo (STEBICEF Department). The animals were maintained for 2 weeks, respecting the photoperiod in different tanks (with the same conditions) containing aerated, filtered, refrigerated seawater (16 ± 1 °C), at a salinity level of 38‰ and with oxygenation of 8 mg/L. During this period, the animals were fed with invertebrate food (Azoo, Taikong Corp., New Taipei, Taiwan) until 24 h prior to the beginning of the experiment (which were performed according to the guidelines of the OECD, 2012 [52]).

### 2.2. Gamete Collection

The emission of the gametes from each individual was induced by the intracelomic injection of 1 mL of 0.5 M KCl through the peristomial membrane [53,54,55]. The eggs and spermatozoa were collected and washed 3 times in filtered seawater (FSW: 0.5 M NaCl, 8 mM KCl, 30 mM Na_2_SO_4_, and 2 mM NaHCO_3_, with a pH 8.1, and pasteurized at 60 °C); the vitality and the number of the gametes was assessed using a Neubauer chamber [56].

### 2.3. Experimental Plan

SMX was dissolved in the FSW to perform three different experiments:-The simultaneous exposure (SE) of the gametes, which were simultaneously combined in multiwell plates with the experimental solutions. The effects were observed on the first embryonic division after 4 h. The percentages of unfertilized eggs and the regular divisions were evaluated;-The pre-exposure (PE), which was achieved by pre-treating the eggs and spermatozoa separately for 2 h before fertilization. The percentages of unfertilized eggs and the regular divisions were evaluated; and-The post-fertilization exposure (PFE), in which previously fertilized eggs were subsequently exposed to differing drug concentrations for 2 h. The effects were observed starting at 4 h post-exposure. The percentages of irregular, regular, and delayed divisions were evaluated.

All experiments were performed in polystyrene, 6-well plates with SMX solutions at final concentrations of 0.05 mg/L, 0.5 mg/L, 5 mg/L, and 50 mg/L (with a 10 mL final volume for each well). Embryotoxicity was evaluated using eggs and sperm at a 1:8 ratio (1 × 10^6^/mL eggs and 8 × 10^6^/mL sperm). Each experiment (SE, PE, and PFE) was repeated 3 times using 6 individuals for replication (18 in total: 9 males and 9 females). For all experiments and for all experimental points, a total of 100 unfertilized eggs (including regular, irregular, and delayed divisions) were counted, and the data were expressed as the number in each individual category.

### 2.4. Statistical Analysis

All of the results obtained were grouped by experimental condition (simultaneous exposure, preventive exposure, and post-fertilization exposure) and concentration of sulfamethoxazole treatment (0.05, 0.5, 5, 50 mg/L) comprising the control group for the statistical analysis. The assumption of the normality of distribution and homogeneity of variance were verified before the analysis. One-way ANOVA and non-parametric Kruskal–Wallis tests were carried out to verify significant differences in unfertilized eggs, regular divisions, and delayed divisions (only in the PFE analysis) between data groups. In the case of significant differences, parametric (Tukey test) and non-parametric (Kruskal–Wallis post-hoc test) multiple comparison tests were performed. The statistical analysis was performed with R 3.2.2 software (R Development Core Team) using the R Commander environment.

## 3. Results

### 3.1. Simultaneous Exposure Effects

The simultaneous exposure (SE) of the gametes to the experimental solutions of the drug showed significant increases in the percentage of unfertilized eggs compared to the controls (2.89 ± 0.76%) only at concentrations of 5 mg/L (12 ± 0.61%) and 50 mg/L (15.10 ± 3%) (Df = 4; F value = 120.8; *p* < 2 × 10 ^− 16^) (Figure 1A).

Regarding the percentage of regular divisions, the results showed a dose-dependent trend. As shown in Figure 1B, compared to the control (92.29 ± 1%), a significant decrease was observed after exposure to 0.5 mg/L, 5 mg/L, and 50 mg/L of SMX (70.11 ± 3.59%, 49.04 ± 5.81%, and 37.99 ± 5.56% respectively).

### 3.2. Preventive Exposure Effects

In the second set of experiments, the gametes were pre-treated for 2 h before fertilization with the four different concentrations of SMX. The exposure of the gametes to the drug showed a dose-dependent effect on fertilization (Kruskal–Wallis chi-square = 49.21, df = 5, *p*-value = 0.000). The multiple comparison after the Kruskal–Wallis test showed that the percentages of the unfertilized eggs resulted significantly higher as compared to the controls (3.06 ± 0.69%) only at two SMX concentrations, i.e., 5 mg/L and 50 mg/L (18.02 ± 1.9% and 29.21 ± 3.69%, respectively; *p* < 0.05) (Figure 2A). The results of the effects on the regular division of embryos demonstrated a significant reduction in the percentage of regular divisions compared to the control samples (mean square = 6358; F value = 302.7; *p* = 2 ×10 ^−16^) at all experimental SMX concentrations (Figure 2B).

### 3.3. Post-Fertilization Exposure Effects

The toxic effects of the exposure to SMX were also evaluated by incubating the embryos at 2 h post-fertilization. Regarding the percentage of regular and irregular divisions, a significant decrease was observed only at two SMX experimental concentrations (5 mg/L and 50 mg/L) compared to the controls (Kruskal–Wallis chi-square = 51.687, df = 5, *p*-value = 6.25 × 10^−10^). In particular, a significant decrease in the percentage of regular divisions (49.3 ± 1.65% and 30 ± 1.66%, respectively) compared to the controls (92.40 ± 0.98%, Figure 3A) and a significant increase in the percentage of irregular divisions (23.43 ± 1.06% and 28.13 ± 1.30%, respectively); Kruskal–Wallis chi-square = 42.429, df = 4, *p*-value = 1.36 × 10^−8^) with respect to the controls (1.97 ± 0.18%, Figure 3B) were observed. On the other hand, the results showed a significant increase in the percentage of delayed divisions of the embryos only at 0.5 mg/L (9 ± 0.23%), 5 mg/L (12.27 ± 0.36%), and 50 mg/L (14 ± 1.67%), as compared to the controls (1.5 ± 0.62%, *p* < 0.05), as showed in Figure 3C.

## 4. Discussion

The growing presence of drugs in the aquatic environment [57], imposes the need to analyze and better understand their impact on aquatic organisms, with particular attention to their reproduction, a critical and important phase of the lifecycle [58]. In fact, in the aquatic environment, most animal species reproduce via external fertilization through the release of gametes into the surrounding environment. When pollutants, including drugs, are poured into the aquatic environment, the gametes are exposed to these substances, which alters their quality and fertilization processes [58]. As reported by Duan et al. [38], although SMX concentrations range from only ng/L to μg/L in natural water, the discharge of livestock and aquaculture wastewater can cause the elevation of SMX levels (54.83 mg/L, [39]). For this reason, with information about the toxicity of this drug being scarce and unclear [38], our study aimed to analyze for the first time its in vitro effects on the gametes and fertilization of *A. lixula* in particular so as to evaluate whether the negative effects occur during the gametes’ simultaneous exposure or during pre- or post-fertilization exposure.

The obtained results show that treatments with different concentrations of SMX negatively affect in a dose-dependent manner the percentage of both unfertilized eggs and the regular divisions of embryos, highlighting the negative effects of the drug on the gametes. In fact, it is known that pollutants present in the aquatic environment can alter the motility of spermatozoa or interfere with the chemotactic mechanisms (spermatozoa’s attraction towards eggs via chemotactic substances) with a consequent reduction in fertility [59]. Moreover, the negative effects observed on the regular divisions could be due to changes in the expressions of genes involved in detoxification, growth, development, and reproduction, or due to the actions of the drug as an endocrine disruptor [60,61]. In light of the results obtained from the simultaneous exposure experiment, we wanted to understand whether the effects of the drug were negative in the case of pre-treating the eggs and spermatozoa for 2 h before fertilization with the different experimental SMX solutions. In this case, as in the case of the simultaneous exposure of the gametes, significant dose-dependent increases in the levels of unfertilized eggs and significant dose-dependent decreases in the levels of regular divisions were observed. To date, no one has analyzed the effects of the pre-exposure to sulfonamides on the gametes of *A. lixula* sea urchins. However, similar results have been reported in studies in which anti-inflammatory drugs, another category of drugs widely detected in wastewater, were tested [62]. In addition, in this case, in our experiment, we could speculate that the pre-exposure of gametes to SMX influences sperm motility, thereby affecting the percentage of fertilized eggs. This is confirmed by Zanuri et al. [63], who investigated the effects of two anti-inflammatory drugs (diclofenac and ibuprofen) on the fertilization of the echinoderms *Asterias rubens* and *Psammechinus miliaris* after pre-incubation of the gametes with the drugs. As discussed above, the reduction in the quality of the gametes, in terms of sperm motility; the inability to fertilize the egg; and the damage to the genetic material could be the main causes of the decreased fertilization rate. Moreover, exposure to the drug could also alter or inhibit the normal mechanisms necessary for fertilization, such as polyspermy prevention and cortical granule exocytosis [64,65], inevitably leading to a reduction in normal embryonic development and subsequently to a reduction in the regular division of embryos.

Once we had evaluated the effects of SMX on the gametes, the third set of experiments in this study allowed us to investigate the consequences of post-fertilization exposure to the different concentrations of SMX on the fertilized eggs. Toxic effects were observed at the highest SMX concentrations, with a significant decrease in the percentage of regular division and an increase in the percentage of irregular and delayed divisions. There is currently no similar evidence in the literature on invertebrates, and particularly on echinoderms, which tests the effects of sulfonamides after fertilization. However, comparable results on the embryos of *Mellita quinquiesperforata* were obtained with anticancer drugs, as shown by the recent study by Mello et al. [66]. Moreover, evidence in the literature reports the anticancer effects of sulfonamides, which exert their action through cell cycle arrest [67,68]. Probably, the concentrations of SMX tested in our study could conceivably block DNA replication, the cell cycle, and cell proliferation. The delays recorded in the development of the embryos could also be caused by the dose-dependent increases in the accumulation of ROS species and the up-regulation of inflammatory genes, which are responsible for alterations and damage at the biochemical and cellular levels [34,69,70]. Furthermore, alterations in the normal development of fertilized eggs, due to interference with the mitotic apparatus could happen, in agreement with a previous study which evaluated the effect of the drug diamino diphenyl sulfone on the embryonic development of the sea urchin *Lytechinus variegatus* [71]. However, there is also conflicting evidence in the literature showing that treatment with 50 mg/L of SMX does not induce obvious alterations during the development of *P. lividus* embryos [72]. In view of this, it is evident that the effects of SMX could be species-specific and could depend on exposure concentrations and combinations with other drugs. It would therefore be important to perform further experiments that clarify the cellular and molecular mechanisms by which SMX negatively affects the embryonic development of *A. lixula*.

## 5. Conclusions

Given the increasingly abundant and disproportionate use of antibiotics that inevitably are poured into wastewater and then into the aquatic environment, it is necessary to investigate their potential toxicological effects. Our analysis on the consequences of the exposure of the gametes and embryos of *Arbacia lixula* to increasing concentrations of the antibiotic sulfamethoxazole (SMX) has shown for the first time in this species the manner in which the evaluated drug affects the reproduction of the sea urchin, causing adverse effects during the simultaneous exposure as well as the pre- and post-fertilization exposure of gametes. This has made it possible to identify that the drug acts negatively from the beginning on the gametes when released into the aquatic environment, reducing the possibility of fertilization. Subsequently, the drug can further negatively affect embryonic development, causing, in the end, negative effects on the fitness of this species. In fact, our results show that the fertilization rate was reduced in a dose-dependent manner, and the fertilized eggs showed alterations in cell division, morphological anomalies, and degeneration. The stages of reproduction are extremely important for animals as they ensure the continuity of the species, and for this reason, such a negative impact of SMX on the fertilization and development processes could lead to a serious reduction in their fitness, endangering the survival of the species and disturbing the entire ecosystem. The choice to use different SMX concentrations in this study allowed us to highlight its toxicity and to determine which concentrations cause the greatest negative effects, and consequently the level of concentration that should not be exceeded. All of this can therefore provide the basis for the proper monitoring and improvement of procedures and technologies suitable for limiting the release of drugs into the aquatic environment.

## Figures and Tables

**Figure 1 animals-12-02483-f001:**
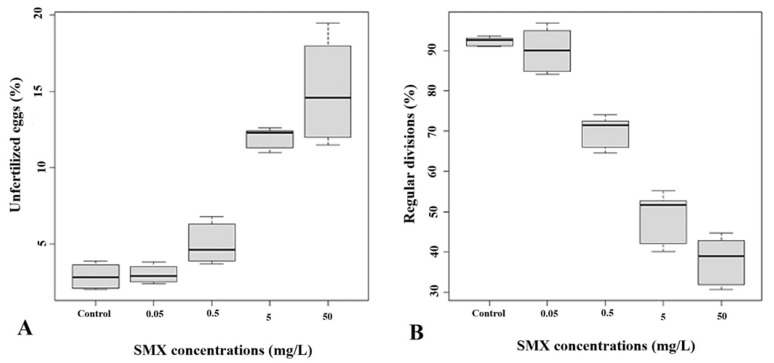
Percentage of unfertilized eggs (**A**) and regular divisions (**B**) considering division stages with a number of blastomeres ≥ 16 after the simultaneous exposure (SE) of the eggs and spermatozoa to the SMX solutions. Black bars represent medians.

**Figure 2 animals-12-02483-f002:**
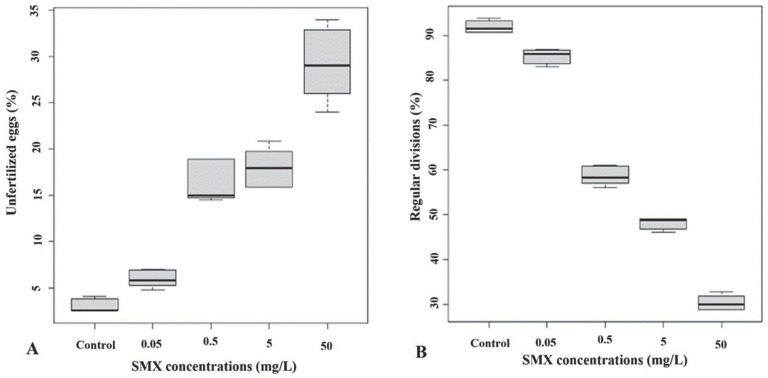
Percentage of unfertilized eggs (**A**) and embryos’ regular divisions (**B**) considering division stages with a number of blastomeres ≥ 16 after preventive exposure (PE) of the gametes to the sulfamethoxazole solutions. Black bars represent medians.

**Figure 3 animals-12-02483-f003:**
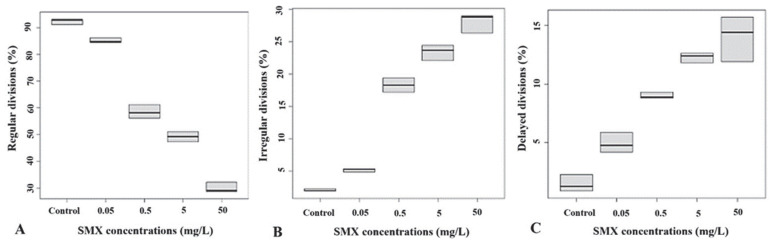
Percentage of embryos’ regular (**A**), irregular (**B**), and delayed divisions (**C**) at 2 h post-fertilization after incubation with sulfamethoxazole solutions (SMX) considering division stages with a number of blastomeres ≥ 16.

## Data Availability

The data presented in this study are available on request from the corresponding authors (mirella.vazzana@unipa.it; vicenzo.ferrantelli@izssicilia.it).

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
