# Peer review of "Effects of Sulfamethoxazole on Fertilization and Embryo Development in the Arbacia lixula Sea Urchin"

_animals, 2022, doi:10.3390/ani12182483_

Round 1

Reviewer 1 Report

General comments:

Although this study proposes interesting data on Sulfamethoxazole toxic effects and emphasises the danger for populations of aquatic organisms, it needs revision before it can be positively evaluated for publication on “animals”. Before going into detail with specific comments, I suggest the authors to present the number of concentrations in increasing order (0.05 mg/L, 0.5 mg/L, 5 mg/L and 50 mg/L), uniform the text accordingly. The figures have a very bad quality, please provide better versions.

Lines 92-127: This section have to be reworked. Why the authors have choosen these specific concentrations, which is the rationale of these solutions? I only see reference about previous studies highlighted sulfamethoxazole toxic effects at certain concentrations.

The experimental design must be improved, in order to do so ,I suggest the authors to take a cue from these papers  https://doi.org/10.3390/toxics10020081https://doi.org/10.3390/toxins13100710 and cite them.  

Author Response

General comments:

Although this study proposes interesting data on Sulfamethoxazole toxic effects and emphasises the danger for populations of aquatic organisms, it needs revision before it can be positively evaluated for publication on “animals”. Before going into detail with specific comments, I suggest the authors to present the number of concentrations in increasing order (0.05 mg/L, 0.5 mg/L, 5 mg/L and 50 mg/L), uniform the text accordingly. The figures have a very bad quality, please provide better versions.

 Answer: Thank you for your suggestion, we revised as you requested. Moreover, we improved the quality of the figures.

Lines 92-127: This section have to be reworked. Why the authors have choosen these specific concentrations, which is the rationale of these solutions? I only see reference about previous studies highlighted sulfamethoxazole toxic effects at certain concentrations.

The experimental design must be improved, in order to do so, I suggest the authors to take a cue from these papers  https://doi.org/10.3390/toxics10020081https://doi.org/10.3390/toxins13100710 and cite them.  

Answer: Thank you for your suggestion, we improved the introduction section adding the references which report the effects and concentrations range SMX of different type of environment. We chose our experimental concentrations studying these references.  We believe that it was more appropriate to motivate these choices in the introduction and discussions sections and not in the materials and methods section.

Reviewer 2 Report

Manuscript describes investigation on the in vitro effects of four concentrations of sulfamethoxazole on the fertilization and development of the sea urchin Arbacia lixula. The gametes were exposed to drugs in three different steps: simultaneously, prior to fertilization and post fertilization.

This problem is very important due to increasing concentrations of drugs especially that they have a serious impact on marine and sweet water creatures. Moreover no one analysed the effects of sulfonamides pre-exposure on the gametes of A. lixula sea urchins up to this study (:) ).

As Authors decibe in lines 269-272 "The choice of a wide range of SMX concentration (0.05 - 50 mg/L) for this study, starting from the concentrations detectable in the environment, allowed to highlight its toxicity and to know which concentrations cause the greatest negative effects, and consequently which concentration is important not to exceed." is bit too high to me. In manuscripts cited by Authors of that paper the highest amount of SAX was 7400 ng/L, which is equal 0.0074 mg/L.  I would recomend for further experiments start from much more lower dose.

Although the manuscript seems to be nicely and meticulously written, the authors made many mistakes related to the incorrect spelling of numbers, due to the fact that in English, after integers and before decimal values, a dot and not a comma are given. That simple detail totally changes the data values (plese check lines 141-142, 145-147, 158-161, 173-179 with the numbers in brakets).

The last comment is about citations - I suggest that "the guidelines of OECD, 1998" (line 100-101) and "Bain et al., 2017" (line 107) should be decribed in References.

Author Response

Manuscript describes investigation on the in vitro effects of four concentrations of sulfamethoxazole on the fertilization and development of the sea urchin Arbacia lixula. The gametes were exposed to drugs in three different steps: simultaneously, prior to fertilization and post fertilization.This problem is very important due to increasing concentrations of drugs especially that they have a serious impact on marine and sweet water creatures. Moreover no one analysed the effects of sulfonamides pre-exposure on the gametes of A. lixula sea urchins up to this study (:) ).

As Authors decibe in lines 269-272 "The choice of a wide range of SMX concentration (0.05 - 50 mg/L) for this study, starting from the concentrations detectable in the environment, allowed to highlight its toxicity and to know which concentrations cause the greatest negative effects, and consequently which concentration is important not to exceed." is bit too high to me. In manuscripts cited by Authors of that paper the highest amount of SAX was 7400 ng/L, which is equal 0.0074 mg/L.  I would recomend for further experiments start from much more lower dose.

Answer: We appreciated your suggestion which we will consider it in our future study and we have moderated the sentence of the conclusions section that you have highlighted to us. However, being that In literature some authors report higher SMX level respect to Blair et al., (2015) "that you cite above" and being in vitro experiments it seemed to us more appropriate use the concentrations range reported in our study.

Although the manuscript seems to be nicely and meticulously written, the authors made many mistakes related to the incorrect spelling of numbers, due to the fact that in English, after integers and before decimal values, a dot and not a comma are given. That simple detail totally changes the data values (plese check lines 141-142, 145-147, 158-161, 173-179 with the numbers in brakets).

Answer: Thank you for your suggestion, we changed comma using dot as you requested.

The last comment is about citations - I suggest that "the guidelines of OECD, 1998" (line 100-101) and "Bain et al., 2017" (line 107) should be decribed in References.

Answer: Thank you, we added the references as you requested.

Reviewer 3 Report

The work is well written and easy to read and follow.
I have just very few comments for the authors.

Write Arbacia, not A. lixula, the first time in the text, line 87.

Did the authors fix the development with formalin or another solution prior to proceed with counting?

Round the p.values at the third decimal unit. Line 156

In some cases, statistical information is lacking. E.g. Line 159-162 and 176-179.

Figure captions. The authors wrote: “Values are expressed as mean ± SD.”. But, if those shown are boxplots, that sentence is wrong. Please correct either the plot or the captions.

Figures. Are the plot showing all the data or only the means of the three experiments? 

Author Response

The work is well written and easy to read and follow.
I have just very few comments for the authors.

Dear Reviewer. Thank you for your important suggestions. We accepted all your suggestions and revised the manuscript as you requested.

Write Arbacia, not A. lixula, the first time in the text, line 87

Answer: Thank you for your suggestion, we changed “A.lixula” in “Arbacia lixula” in line 87.

Did the authors fix the development with formalin or another solution prior to proceed with counting?

Answer: We did not use formalin or another solution because experimental design allowed us to carry out the observations and counts immediately after the incubation time ending.

Round the p.values at the third decimal unit. Line 156

Answer: Thank you for your suggestion, we modified the decimal units as you requested.

In some cases, statistical information is lacking. E.g. Line 159-162 and 176-179.

Answer: Thank you we added the information that you requested.

Figure captions. The authors wrote: “Values are expressed as mean ± SD.”. But, if those shown are boxplots, that sentence is wrong. Please correct either the plot or the captions.

Answer: Thank you for your suggestion, we changed the captions. Sorry for this mistake.

Figures. Are the plot showing all the data or only the means of the three experiments? 

Answer: The plot shows all the data obtained.